# Unraveling Spatial-Spectral Dynamics of Speech Categorization Speed Using Convolutional Neural Networks

**DOI:** 10.3390/brainsci13010075

**Published:** 2022-12-30

**Authors:** Kazi Ashraf Moinuddin, Felix Havugimana, Rakib Al-Fahad, Gavin M. Bidelman, Mohammed Yeasin

**Affiliations:** 1Department of EECE, University of Memphis, Memphis, TN 38152, USA; 2Department of Speech, Language and Hearing Sciences, Indiana University, Bloomington, IN 47408, USA

**Keywords:** categorical perception, behavioral response, frequency bands, convolutional neural network, guided GradCAM

## Abstract

The process of categorizing sounds into distinct phonetic categories is known as categorical perception (CP). Response times (RTs) provide a measure of perceptual difficulty during labeling decisions (i.e., categorization). The RT is quasi-stochastic in nature due to individuality and variations in perceptual tasks. To identify the source of RT variation in CP, we have built models to decode the brain regions and frequency bands driving fast, medium and slow response decision speeds. In particular, we implemented a parameter optimized convolutional neural network (CNN) to classify listeners’ behavioral RTs from their neural EEG data. We adopted visual interpretation of model response using Guided-GradCAM to identify spatial-spectral correlates of RT. Our framework includes (but is not limited to): (i) a data augmentation technique designed to reduce noise and control the overall variance of EEG dataset; (ii) bandpower topomaps to learn the spatial-spectral representation using CNN; (iii) large-scale Bayesian hyper-parameter optimization to find best performing CNN model; (iv) ANOVA and posthoc analysis on Guided-GradCAM activation values to measure the effect of neural regions and frequency bands on behavioral responses. Using this framework, we observe that α−β (10–20 Hz) activity over left frontal, right prefrontal/frontal, and right cerebellar regions are correlated with RT variation. Our results indicate that attention, template matching, temporal prediction of acoustics, motor control, and decision uncertainty are the most probable factors in RT variation.

## 1. Introduction

Categorical perception (CP) of audio is the process of grouping sounds into categories based on acoustical properties [1]. The neurological basis of CP is present in infancy and evolves through auditory training [2]. Thus, CP is elemental in the comprehension of sounds and is a measure of fluency in speech perception. Decoding the neural organization of CP might be important for understanding certain disorders which impair sound-to-meaning associations including age-related hearing loss [3], deafness [4] and dyslexia [5]. To date, various neuroimaging studies have localized brain regions sensitive to acoustical changes in phonemes [6,7,8,9,10]. However, the neural basis of behavioral outcomes in CP is poorly understood. Response time (RT) is one such behavioral metric that measures the speed of identifying phonetic categories by listeners. In this study, we conducted a data-driven investigation to identify the neural underpinnings as well as their respective functional role in eliciting speeded categorization decisions.

In speech perception studies, RT is treated as an indicator of perceptual difficulty because of the direct correlation with the acoustical properties of the stimuli [11,12,13]. For example, faster responses are observed in acoustically identical and large acoustic boundaries whereas the acoustic difference in the same phonetic category causes ambiguity and slower responses [11]. Aside from measuring perceptual difficulty, RTs are also correlated with decision difficulty in listening tasks. RTs increase with the number of decision choices [13] and can be considered a metric for measuring cognitive load during speech perception. Despite being widely used as a measure of perceptual difficulty, the sources of RT variations and in association with different neural regions need further investigation. We identified two factors that may be relevant to understand the sources of RT variation in CP: (i) the role of regions inside the CP categorization network hub and their relative importance for determining listeners’ decision speed, (ii) the effect of right hemispheric regions on speeded categorization.

Identifying the effect of regions inside the auditory processing circuitry would clarify how different brain processes affect decision speed in CP. Although speech categorization is dominant in the left hemisphere, studies have also shown the involvement of right hemispheric regions during CP tasks [14]. For instance, the right hemisphere is engaged in processing lexical tones [15,16,17], acoustic information such as pitch levels [18] and processing of non-speech stimuli such as musical tones [19,20]. So, it is important to decode the association of right hemispheric regions and RT to fully understand the neural basis of RT variation in CP. In this study, we aimed to characterize left and right hemispheric factors from objective measures of neuronal function without any prior assumptions.

In addition, frequency band analysis of EEG/MEG signals has revealed different cognitive functions associated with CP that are carried in different oscillatory brain rhythms. In particular, oscillations in α (9–13 Hz), β (14–30 Hz) and γ (>30 Hz) bands are linked with attention [21], template matching [22] and network synchronization [23,24,25] respectively. Establishing the spectral characteristics of brain regions would further aid in understanding their respective functional role in speech perception and categorization. To this end, we assessed how different oscillatory characteristics in α,β,γ frequency bands map to behavioral RTs to elaborate their functional role in rapid speech categorization.

We conducted our analysis on EEG data acquired from a CP experiment where participants were tasked to identify two phonetic categories (‘oo’ or ‘aa’) from 5 equidistant vowel tokens (/u/ to /a/) [26,27,28]. We built a unified framework utilizing learned representation from deep learning models and statistical analysis to identify neural correlates of RT. In this framework, we have incorporated a data augmentation algorithm, spatial-spectral representation, parameter-optimized modeling using convolutional neural networks (CNNs), and class discriminative visualization (Guided-GradCAM) for model interpretation. The framework can be summarized in four steps:1.Data augmentation: To robustly model cognitive events from EEG data using deep learning (DL) tools, it is necessary to address two constraints of EEG datasets. The first is the issue of noise prevalence in EEG data and the second is the small sample size problem of EEG datasets. We have addressed these issues by adopting a data augmentation process [29] for generating event-related potentials (ERP) from EEG samples. The algorithm is designed to reduce noise as well as control the overall variance of the dataset for robust modeling.2.Spatial-spectral representation: Bandpower features are one of the effective ways to capture spatial-spectral properties of EEG data. We extend the bandpower features to have an image representation, this is done so to include the specific location of the neural regions during modeling.3.Large scale parameter optimized model: We used CNN to model the underlying spatial-spectral properties of RT. CNN is known for its effective spatial modeling and has proven performance in modeling cognitive events from EEG data. We deployed the Bayesian hyperparameter optimization algorithm, tree-structured Parzen Estimator (TPE) [30] to find the best configuration for our CNN model.4.Model interpretation: To discover the spatial-spectral correlates of RT, we dive into the learned representation of the CNN models. Specifically, we combined high-resolution visual interpretation techniques like Guided-GradCAM [31] and statistical analysis to discover the underlying factors of RT.

Similar to Al-Fahad et al. [32], we first formed clusters within the RT distribution to capture unique neural patterns confined to the range of fast, medium and slow response speeds. We generated the augmented ERPs within the RT categories of each subject to ensure that individual variations are not nullified. From the augmented ERPs, we extract α,β,γ bandpowers and transform these features into images using bandpower topomaps. Next, we use CNN to learn the spatial-spectral patterns of each category from the bandpower topomaps and use Guided-GradCAM [31] to provide insight into the learned representation of the model. Guided-GradCAM is a high-resolution class discriminative mapping technique that allows visual depiction of the learned feature importance of CNN models [31]. We extract α,β,γ activation values of each neural region from the Guided-GradCAM feature maps to quantify the learned feature importance by the CNN model. Further statistical analysis of these activation values was carried out to unravel the spatial-spectral correlates of RT.

Our empirical data and computational model reveal how different spatial-spectral correlates map variations in speech CP. The significant effect of α−β bands suggests that attention and template matching are major factors driving decision speeds, while the involvement of right prefrontal/frontal and cerebellar regions indicate motor control, decision uncertainty and temporal prediction of acoustics are other factors in RT elicitation. Overall, our study incorporates novel analysis applied to EEG data to uncover the neural basis of RT variation as well as validate prior findings of brain-behavior relationships in auditory CP.

## 2. Methodology

### 2.1. Participants

We conducted our analysis on previous published data from several CP experiments [26,27,28]. N = 15 males and 35 females aged 18 to 60 years originally participated. All of the participants were recruited from the University of Memphis student body and the Greater Memphis area. Participants had normal hearing sensitivity (i.e., <25 dB HL between 250–8000 Hz) and were strongly right-handed (mean Edinburgh Hand Score ≈ 80.0%). Participants had a range of musical training varying from 1 to 27 years. All participants were paid for their time and gave informed consent in compliance with the Institutional Review Board (IRB) at the University of Memphis.

### 2.2. EEG Recording & Preprocessing

During the experiment, the participants were instructed to listen to sounds from a five-step vowel continuum; each token of the continuum was separated by equidistant steps based on the first formant frequency (F1) categorically perceived as /u/ to /a/. Each token was 100 ms long, including 10 ms rise and fall time. The stimuli were delivered through shielded insert earphones; listeners heard 150–200 trials of individual tokens and were asked to label the sound as perceived through binary responses (‘u’ or ‘a’). Response times (RTs) were recorded as the difference between the stimulus onset and the behavioral response (i.e., labeling of the token). Simultaneous EEGs recording were recorded using 64 channels sintered Ag/AgCI at standard 10–10 electrode locations around the scalp. As subsequent preprocessing steps, ocular artifacts were corrected using principal component analysis (PCA), filtered (bandpass: 1–100 Hz; notch filter: 60 Hz), epoched (−200 to 800 ms) into single trials, and baseline corrected (−200 ms to 0 ms) [26,27,28].

### 2.3. Clustering RTs

RTs are a continuous variable indexing perceptual response speed. For data reduction purposes, we clustered (in an unsupervised manner) the RT data into groups describing fast, medium, and slow speeds. Al-Fahad et al. followed a similar approach and was able to decode functional connectivity patterns unique to these three categories of RT [32]. Thus, to find unique spatial-spectral patterns associated with different orders of RT, we first formed clusters within the RT distribution in an unsupervised manner. We identified four clusters in our RT distribution through Gaussian Mixture Model (GMM) with the Expectation Maximization (EM) algorithm. Out of four clusters, three represented fast, medium and slow RTs while the other cluster was deemed an outlier due to its low probability. The optimal GMM model (number of components = 4, covariance type: spherical) was selected by comparing the Bayesian Information Criterion (BIC) scores among a finite set of models. Figure 1A shows the BIC scores of GMMs with different configurations. Among the selected three clusters, the majority of responses in our data were fast followed by medium and slow trials. The uneven distribution of samples across these clusters necessitated the use of a bootstrapping process.

### 2.4. Bootstrap and Eigenspace Filtering

We computed event-related potentials (ERPs) by averaging a small number of trials. Since DL models require a high number of input samples for training, we bootstrapped the ERPs as a data augmentation approach. Aside from augmentation, we realized the need to use a more controlled bootstrap sampling algorithm to balance the noise and the overall variance across samples (see Algorithm 1). The main goal of this strategy is to augment our dataset so that each sample is less noisy than the original EEG while also lowering the total variance of the dataset but not completely biasing it. To begin, we sampled a small number of trials (5∼6%) to average and then repeated the process for a large number of iterations. Additionally, we use category-specific dropout rates during the sampling and an eigenspace sample filtering criterion as a bias reduction mechanism for limiting the number of identical samples.
**Algorithm 1** bootstrap ($X,r,\alpha ,n,n_{\alpha },\theta$)X→EEGtrialsr→no.oftrialstoaverage$\alpha \to [\alpha_{1},\alpha_{2},\ldots ,\alpha_{c}]\;\mathrm{category}\;\mathrm{specific}\;\mathrm{dropout}\;\mathrm{rates}$n→no.ofiterations$n_{\alpha }\to \mathrm{predetermined}\;\mathrm{interval}\;\mathrm{to}\;\mathrm{drop}\;\mathrm{trials}$$\theta \to [\theta_{1},\theta_{2},\ldots ,\theta_{c}]\;\;\mathrm{categories}$$\hat{X}\leftarrow \left[\;\right]$**for**$\theta_{i}$ in θ **do**    $x_{\theta_{i}}\leftarrow X\left[\theta_{i}\right]$    $m_{i}\leftarrow r\times \alpha_{i}$    **for** j=1 to *n* **do**          **if** $|x_{\theta_{i}}|\leq r$ **then**                $\hat{X}\left[j\right]\leftarrow \frac{1}{|x_{\theta_{i}}|}\sum_{x_{k}\in x_{\theta_{i}}}x_{k}$        **end if**        $\hat{x}\leftarrow sample\left(x_{\theta_{i}},r\right)$                  ▹ sample *r* trials from $x_{\theta_{i}}$        $\hat{X}\left[j\right]\leftarrow \frac{1}{r}\sum_{k=0}^{r}{\hat{x}}_{k}$        **if** $j\;\mathrm{rem}\;n_{\alpha }=0$ **then**               $x_{m}=sample\left(\hat{x},m\right)$                  ▹ select $m_{i}$ trials to drop               $x_{\theta_{i}}=drop\left(x_{\theta_{i}},x_{m}\right)$                  ▹ drop selected $x_{m}$ trials        **end if**        **if** $x_{\theta_{i}}=⌀$ **then**               break        **end if**    **end for****end for**

#### 2.4.1. Category Specific Dropout

We grouped trials of each subject according to their RT categories (fast, medium, slow) and applied the aforementioned bootstrapping process within each category of individual subjects. During the sampling process, we dropped trials randomly to avoid generating similar ERPs. The trials were dropped according to a dropout rate parameter which is specified in prior for each RT category separately. The dropout rate for an RT category was specified based on the sample size of that category. This controlled the variance and the number of observations in each RT category. Based on the category-specific dropout rates, the algorithm produced different numbers of samples. If the bootstrap algorithm is run for *n* iterations for any ith category $\theta_{i}$ and the predefined interval is $n_{\alpha }$, then trials are dropped $\frac{n}{n_{\alpha }}$ times in total. Accordingly, for any sampling rate *r* and $\alpha_{i}$ as the dropout rate of $\theta_{i}$, the total number of samples dropped is $d_{i}=\frac{n}{n_{\alpha }}\times r$. If *f* is the function that outputs the number of samples generated by the bootstrap algorithm for $\theta_{i}$, then,
$$f\left(x_{\theta_{i}}\right)=\left\{\begin{array}{c} 1\;\mathrm{if}\;|x_{\theta_{i}}|<r\\ n\;\mathrm{if}\;|x_{\theta_{i}}|\geq n\\ \frac{|x_{\theta_{i}}|n_{\alpha }}{r}\;\mathrm{if}\;|x_{\theta_{i}}|<n \end{array}\right.$$

We had a total of 45,550 single trials (slow: 4025, med: 11,029, fast: 30,496) from 50 participants. After applying the bootstrap algorithm with parameters $r=50,\;\alpha =(\alpha_{slow}=0.1,\alpha_{med}=0.2,\alpha_{fast}=0.3),\;n=1500$ and $n_{\alpha }=10$, we achieved 133698 number of samples (slow: 28,550, med: 46,368, fast: 58,780). The proportion of samples in slow, medium, and fast RTs was altered from 9% to 21%, 24% to 35%, and 67% to 44% respectively. Even with dropouts, it is possible to generate identical samples. To further reduce the bias of our generated samples, we used distance-based eigenspace filtering.

#### 2.4.2. Eigenspace Filtering

The purpose of eigenspace filtering is to drop samples that are too far or close to the raw EEG samples. We created an eigenspace of the raw EEG samples through Principal Component Analysis (PCA) and projected the augmented samples in that space. Next, the augmented samples were reconstructed with all the components and the reconstruction error of each sample was observed. Figure 2 shows the distribution of reconstruction error of the augmented samples.

The reconstruction error was distance measures from the EEG eigenspace and constitutes how far the augmented samples are from the raw EEG data. Samples that fell within −3σ to −2σ of the mean error were biased samples and therefore were eliminated to reduce the overall bias of the data. Similarly, samples within +2σ to +3σ are noisy due to their similarity with the raw EEG data and are removed to control the variance. In short, we retained samples between the 25th–75th percentiles of the error distribution which in effect reduced the size of our data by 50% while still containing enough variation to learn the underlying factors of RT.

### 2.5. Spatial-Spectral Representation

Standard approach in EEG analysis is to use spectral contents like bandpower signals in the form of 1-dimensional feature vectors. However, Bashivan et al. pointed out that representing bandpowers into feature vector forms cannot be considered a spatial representation due to violation of the inherent structure of data in space [33]. We are required to include the electrode locations in addition to their spectral content in order to decode neural regions driving RT in CP. To this extent, we used an image representation of the bandpower signals in which the location of each electrode and its frequency content is accounted for. The image representation used a topographic structure of the scalp where each electrode was placed in the structure by projecting their original 3D coordinates to 2D surface [33]. The bandpowers of each electrode were then mapped into a topographic surface and interpolated to create a spatial representation called bandpower topomaps (Figure 3). The composite topomaps allowed us to represent the activation of neural regions as well as their spectral contents through a single representation.

To generate these topomaps, we computed the power spectral density of each ERP sample across frequencies ranging from 0 to 60 Hz. Next, we calculated the bandpowers of α (9–13 Hz), β (14–30 Hz) and γ (31–60 Hz) frequency bands. These bandpower signals were then projected by individual bands into the topographic representation. In this process, each ERP signal is transformed into three grayscale topomaps representing α, β and γ frequency bands. These α, β, and γ topomaps are then stacked in RGB channels to create a spatial-spectral representation of the ERPs [33]. We implemented the bandpower computation and the subsequent topomap creation process using the python library mne [34]. Note that our representation does not include a temporal representation since spectral data were computed across the entire trial epoch window.

### 2.6. Modeling

We used a CNN to model the categorical RTs from bandpower topomaps. We used Bayesian Hyperparamter Optimization with a Tree-Structured Parzen Estimator [30] to find the optimal hyperparameters for our model from a large hyperspace. TPE is an algorithm in the framework of Sequential Model-Based Optimization (SMBO) [35] which involves optimizing the surrogate of a costly fitness function. TPE achieves this numerical optimization by distinguishing points or hyperparameters with low and high probability densities.

The design of our fitness function involved tweaking the architecture of the CNN models as well as a general set of hyperparameters such as learning rate, optimizers, and regularization parameters. We ran the TPE algorithm for 73 trials and evaluated the trials based on validation accuracy. The best model architecture as optimized by TPE contains 4 convolution and 4 inception reduction layers [36,37] with 2 fully connected (FC) layers followed by the softmax layer (Figure 4). The optimizer and learning rate chosen was Adagrad [38] and 0.006 respectively. Model was trained for 165 epochs with 25% of the data used as a validation set. Options to save the model with the best validation loss and reduction of learning rate at plateau were enabled. Model configuration was implemented using the TensorFlow functional API Keras [39] and we used the hyperopt [40] library for TPE optimization.

### 2.7. Band Specific Class Activation Maps

To help interpret the learned representations of these deep neural models, we used Guided GradCAM [31], a class discriminative visualization technique to visualize the spatial activations for different RTs across separate frequency bands (right of Figure 5). Guided GradCAM combines high-resolution Guided Backpropagation [42] pixel-space gradient map with class discriminative map of GradCAM [31] to highlight class-associated feature importance. We applied Guided GradCAM to the test samples and acquired the learned spatial representation in each frequency band. Figure 5 illustrates some examples of the band-specific localization of RTs. We extract activation values from the 64 spatial locations of the saliency maps to further analyze the spatial-spectral factors contributing to variation in RT. The purpose of our analysis was to determine whether and what differences in activation trends exist between RT groups.

### 2.8. Statistical Analysis

To assess how Guided-GradCAM activations (i.e., feature scores) vary with behavioral RTs we performed a mixed-effect ANOVA. The model included factors of electrode (64 levels), frequency band (3 levels), and predicted RT groups (3 levels). We used Tukey HSD posthoc tests to contrast activation in electrodes and frequency bands between the RT categories. Statistical analysis was performed in R (lmer4 [43] and lmerTest [43] packages).

## 3. Results

### 3.1. Model Performance

We evaluated our CNN model performance on a test dataset that the model had never seen. Among the 73 CNN models generated by the TPE algorithm, the mean test accuracy was 56.62%, with the best-performing model achieving a test accuracy of 70.8%. We selected the best model for further analysis. Our model showed an average (macro) of 0.71 precision, 0.7 recall, and 0.71 f1-score. The slow, fast, and medium RTs achieved an average softmax score of 0.84, 0.84, and 0.8 respectively. Table 1 shows the details of model performance. Since there was an imbalance in sample size in the RT categories (slow:fast:medium ≈ 19:43:38), we used precision-recall metrics to diagnose the model. The right of Figure 6 shows the precision-recall trade-off under different decision thresholds. We tested the model performance on each RT category by comparing their average precision (AP) scores. According to the AP scores, our model showed better performance in classifying fast RTs. This might be due to their over-representation in the data and less uncertainty in terms of spatial-spectral patterns. In contrast, the medium RTs were classified less accurately (AP = 0.76) than their counterparts.

The randomness introduced by the bootstrap data generation process makes it difficult for us to confidently identify the specific reason for this performance degradation. For example, the sampling process can shift the mean of RT distributions because of the random dropout of trials, so medium RTs could contain a fair amount of samples near fast and slow RTs, therefore, are classified as such by the model. It could also mean that variation in motor execution speed plays a role and sometimes RTs reflect early or late execution by the participants rather than the speed of their speech categorization process.

### 3.2. ANOVA Results

The mixed-model ANOVA revealed significant variation in activation across electrode sites [F (63, 13,637) = 7.49, *p* < 2.2 $\times \;10^{-16}$], frequency bands [F (2, 13,632) = 33.38, *p* = 3.43 $\times \;10^{-15}$] and RT groups [F (2, 13,635) = 22.93, *p* = 1.14 $\times \;10^{-10}$]. We also found a significant interaction in activation between electrode × band [F (126, 13,637) = 19.49, *p* < 2.2 $\times \;10^{-16}$], electrode × RT [F (126, 13,633) = 1.63, *p* = 1.20 $\times \;10^{-5}$], band × RT [F (4, 13,631) = 2.71, *p* < 2.2 $\times \;10^{-16}$] and electrode × band-RT [F (252, 13,633) = 1.58, *p* = 1.47 $\times \;10^{-8}$]. The 3-way interaction indicates that CNN activations varied with unique spatial-spectral patterns dependent on listeners’ perceptual RTs. Posthoc analysis was conducted to parse this complex interaction and identify possible differences between left vs. right hemisphere data in driving RT behaviors.

For the left hemisphere, increased left frontal activation (FT7, FC5, FC3) was associated with faster responses ($M_{fast}$ = 1.076, $M_{med}$ = 0.79, $M_{slow}$ = 0.27), suggesting an inverse relationship between brain activation and behavior. We also found that temporal and central regions (TP7, C5, C1) were among the other left hemispheric regions distinguishing listeners’ RTs. Activation at TP7 differed between fast-medium [*p* = 0.0036, $\left(M_{fast},M_{med}\right)$ = (1.998, 4.746), z ratio = −3.222] and medium-slow [*p* = 0.0640, $\left(M_{med},M_{slow}\right)$ = (4.746, 2.760), z ratio = 2.24] RTs but not for fast-slow [*p* = 0.643, $\left(M_{fast},M_{slow}\right)$ = (1.998, 2.76), z ratio = 2.24] RTs. Similarly, activation patterns differed in the C5 region but only between medium and slow RTs [*p* = 0.0258, $\left(M_{med},M_{slow}\right)$ = (4.413, 2.368), z ratio = 2.593]. The linear relation between frontal (but not temporal or central) regional activation and RTs suggests that left frontal activity is a major driver of listeners’ perceptual decision speed during speech categorization.

For the right hemisphere, superior prefrontal regions (AF4, F2) contrasted some RT categories. Activation at AF4 contrasted fast-medium [*p* = 0.0087, $\left(M_{fast},M_{med}\right)$ = (3.511, 5.903), z ratio = −2.958] and medium-slow [*p* = 0.0002, $\left(M_{med},M_{slow}\right)$ = (5.903, 2.69), z ratio = 3.948] but no significant contrast between fast-slow [*p* = 0.56, $\left(M_{fast},M_{slow}\right)$ = (3.511, 2.69), z ratio = 0.82] RTs. Similarly, activation in the F2 location contrast between fast-med [*p* = 0.0196, $\left(M_{fast},M_{med}\right)$ = (2.626, 4.945), z ratio = −2.69] and med-slow [*p* = 0.0001, $\left(M_{med},M_{slow}\right)$ = (4.945, 1.255), z ratio = 4.124] but not fast-slow [*p* = 0.27, $\left(M_{fast},M_{slow}\right)$ = (2.626, 1.255), z ratio = 0.82] RTs. These results indicate that high activation in the right prefrontal/frontal regions causes minor decay in RT and transition from fast to medium RTs. Additionally, a right cerebellar region (CB2) distinguished fast-slow [*p* < 0.0001, $\left(M_{fast},M_{slow}\right)$ = (5.055, 8.599), z ratio = −4.351] and med-slow [*p* < 0.0001, $\left(M_{med},M_{slow}\right)$ = (4.320, 8.599), z ratio = −5.253] but not fast-med [*p* = 0.63, $\left(M_{fast},M_{med}\right)$ = (5.055, 4.320), z ratio = 0.919] RTs. The comparatively higher activation at CB2 in slower RTs perhaps indicates that right cerebellar activity is a major cause for late decision speed during CP. Figure 7 depicts the activation ranking of regions localized to each RT category by the CNN model.

### 3.3. Spectral Correlates of RT

To establish the role of different frequency bands on behavioral RT, we next compared average α, β, and γ activation between the RT categories (Figure 8). We found β oscillations ($M_{\beta }$ = 3.12) encodes RT variation most significantly followed by α oscillations ($M_{\alpha }$ = 3.06). Posthoc analysis showed that α activation differed between fast-slow [*p*-value = 0.0287, $\left(M_{fast},M_{slow}\right)$ = (3.03, 2.52), z ratio = 2.554] and med-slow [*p*-value < 0.0001, $\left(M_{med},M_{slow}\right)$ = (3.38, 2.52), z ratio = 4.399] RTs but not fast-med [*p*-value = 0.1577, $\left(M_{fast},M_{med}\right)$ = (3.03, 3.38), z ratio = −1.837] RTs. In contrast, β activation differed between fast-med [*p*-value = 0.0074, $\left(M_{fast},M_{med}\right)$ = (3.04, 3.64), z ratio = −3.01], fast-slow [*p*-value = 0.008, $\left(M_{fast},M_{slow}\right)$ = (3.04, 2.52), z ratio = 2.985] and med-slow [*p*-value < 0.0001, $\left(M_{med},M_{slow}\right)$ = (3.64, 2.45), z ratio = 5.976] RTs. We found γ band to be insignificant in dictating RT variation as posthoc analysis showed no significant contrasts in activation scores between the RT categories (*p* > 0.3).

Considering electrode location alongside frequency bands provided a more specific look at these effects. We found that higher α activation was largely constrained to the left hemispheric regions whereas higher β activation was associated with right hemispheric regions. We observed that the linear activation trend in the left frontal regions is associated with α activation. Our results suggest that RTs get faster as left frontal α activation increases ($M_{fast}^{\alpha }$ = 18.15, $M_{med}^{\alpha }$ = 18.22, $M_{slow}^{\alpha }$ = 13.39). The other source of α activation is the CP1 electrode. Results showed that α activation in this region differs slightly between fast-med [*p*-value = 0.0624, $\left(M_{fast},M_{med}\right)$ = (11.81, 8.77), z ratio = 2.255], med-slow [*p*-value = 0.1571, $\left(M_{fast},M_{med}\right)$ = (11.81, 8.77), z ratio = 1.838] and largely between fast-slow [*p*-value = 0.0002, $\left(M_{fast},M_{slow}\right)$ = (11.81, 6.34), z ratio = 3.999] RTs.

Our analysis revealed that the majority of β activity associated with RT variation is right hemispheric. We find significant contrast of β activation in the right frontal/prefrontal areas. To start, we see that β activation in AF4 electrode between fast-med [*p* < 0.0001, $\left(M_{fast},M_{med}\right)$ = (3.78, 10.07), z ratio = −5.349] and med-slow [*p* < 0.0001, $\left(M_{med},M_{slow}\right)$ = (10.07, 5.43), z ratio = 4.081]. We see a similar β activation trend associated with F2 location, the β activation in this location contrasts between fast-med [*p* < 0.0001, $\left(M_{fast},M_{med}\right)$ = (6.23, 13.61), z ratio = −5.423], med-slow [*p* < 0.0001, $\left(M_{med},M_{slow}\right)$ = (13.61, 2.51), z ratio = −5.423]. These results suggest that right frontal/prefrontal β activity encodes a similar minor transitional effect on RT (fast → med) like left temporal and central regions.

In addition to right prefrontal/frontal regions, the result of our analysis also suggests that right cerebellar β activity is a major factor in determining response speed. From the posthoc analysis, we observe that right cerebellar β activation significantly differ between fast-slow [*p* < 0.0001, $\left(M_{fast},M_{slow}\right)$ = (9.76, 16.42), z ratio = −5.382], med-slow [*p* < 0.0001, $\left(M_{med},M_{slow}\right)$ = (7.19, 16.42), z ratio = −9.22]. The right cerebellar β activity seemed to contain a sub-linear relationship with RTs which implies that an increase in β activity in these regions causes the response to decay gradually. Figure 9 illustrates band-wise ranked regional activation for the respective RT categories.

## 4. Discussion

We have conducted an in-depth spatial-spectral analysis of EEG data along with computational modeling to answer two questions regarding the speed with which human listeners categorize speech: (1) What are the effects of neural regions in inducing RTs during CP? (2) How do right hemispheric regions associate with RT variation?

In this section, we first summarize and discuss the findings regarding each of these questions. Aside from exploring these unknowns, we contributed to developing a novel DL-based approach to decode neural functionalities from EEG data. To the best of our knowledge, this is the first computational EEG decoding framework using CNNs that allows the identification of spatial-spectral correlates of cognitive events. Our proposed approach ensures a fully data-driven procedure without the effects of hand-engineered features and prior assumptions. We present further arguments for our choice of using CNN to model and interpret neural factors from EEG data. Finally, we offer limitations of our study and present viable explorations to further the understanding of brain-behavior relations in speech perception.

### 4.1. Effects of Neural Regions on Categorization Speed

Our first aim was to establish the effect of left hemispheric frontal-temporal-parietal regions on RT variation. These regions are associated with various aspects of the categorization process [32,44,45] and can be thought of as the CP circuit of the human brain. Consequently, measuring the effect of the individual regions inside this network can provide insight into the exact manner in which categorization processes affect RT during speech perception. Here, we show left frontal and central regions are the primary of RTs inside the CP hub. We observe monotonically increasing α activation in these regions as responses get faster and consequently decay as α activation is suppressed. One possible source of the activation contrast in left frontal electrodes FT7, FC5, and FC3 could be the left inferior gyrus (IFG). Left IFG is of course engaged in a wide variety of speech and language functions. IFG is a known marker of phonetic categorization capacity [46], which aligns with our data that left frontal regions drive individuals’ phonetic categorization, at least with respect to decision speed. Still, the dominant left frontal α activation for faster responses perhaps points towards a simpler explanation. Dimitrijevic et al. found positive correlation effect between α activation in the left IFG and listening effort in speech-in-noise conditions [47]. As α oscillations are related to attention processes, it is plausible that left frontal α activation in our analysis reflects the participants’ listening effort.

The C5, C1 and CP1 electrodes cover posterior frontal regions, approximately near the primary motor cortex (PMC). Evidence suggests that both motor and non-motor representations are part of the categorization process [48,49,50]. Still, the exact nature of how putative motor actions relate to speech perception is beyond the scope of our study. Thus, we consider an alternate cause for the saliency of the PMC. Since motor execution time is also part of RT, we consider these sparse activations across the left central regions a reflection of varying motor execution speeds. α activation at CP1 followed a linear trend with RT suggesting faster responses with increasing α activity. Moreover, the leftward lateralization of this motor response is perhaps expected given our sample was predominantly right-handed.

We found little evidence that the auditory cortex played a key role in eliciting RT. TP7 resides over the left auditory temporal cortex. Since activation patterns in TP7 did not differ much between fast and slow RTs, we can assume that neural operations in the auditory cortex have a comparatively lower influence on the speed of listeners’ responses (i.e., RTs) compared to frontal regions. From this observation, we infer that stages of encoding speech signals have a minor effect, and it is the later processes that are the prime determinant of RT. The reasoning of this inference is that regions in the auditory cortex such as the primary auditory cortex (PAC) are responsible for the encoding of speech signals [51,52]. Additionally, we found that γ band is a minor correlate of RT and γ band is important in describing stimulus encoding processes of CP, but not necessary response selection (cf. RT), per se. The minor role of the temporal cortex and the major role of frontal regions is supported by the work of Binder et al. who also showed (with fMRI) that response selection is driven by the left inferior frontal lobe [53].

### 4.2. Right Hemispheric Effect on Categorization Speed

Right hemisphere brain regions are associated with a plethora of auditory (and non-auditory) processes including memory [54], attention [55], decision [56] and motor control [57]. These are all plausible functions that might affect reaction and decision speed during speech categorization tasks. Our results show that the right frontal/prefrontal and cerebellum are important regions that affect RTs in a non-linear fashion. We find that right frontal/prefrontal β activity encodes immediate transitions from fast to medium RTs. It is possible that β activity in these regions causes a slight delay in listeners’ responses due to increased inhibitory motor control [57,58]. In a broader sense, such delays in motor function might be equally described in terms of decision uncertainty. The other plausible theory is that right frontal/prefrontal β activity reflects delay in recall processes. We consider this possibility because right prefrontal regions are associated with memory retrieval operations during speech perception [54]. Regions such as the dorsolateral prefrontal cortex (DLPFC) are associated with working memory [59] and one of the markers for higher auditory perceptual skills is better working memory [60].

Addition to the right frontal areas, we also found increased right cerebellar β activity to hinder RT speeds. Our analysis shows large contrast in right cerebellar β activation between fast and slow trials. Thus, it is apparent that cerebellar β oscillations are significant predictors of early or late responses when categorizing speech stimuli. The right cerebellum relates to linguistic functions such as motor speech planning, verbal working memory, syntax processing and language production [61]. One characterization of the cerebellum is as an ‘internal clock’ [62]. Based on this hypothesis, studies have concluded that the cerebellum is responsible for processing temporal structure of acoustic signals [63]. Such a clocking function could play a role in driving listeners to produce early vs. late responses in our speech perception task. Studies also suggest the cerebellum is tied to language training and verbal working memory [64]. On this account, it is possible that language experience is reflected in cerebellar activity and decision speeds in speech perception are modulated by such experience. However, the most probable theory points toward temporal prediction of input acoustics. Ruiz et al. showed that β oscillation in the cerebellum plays a major role in sequence prediction during speech perception and vocalization [65]. Furthermore, their study also provided evidence suggesting that β oscillations reflect a reconfiguration of sensorimotor representations due to wrong predictions. Consequently, the incorporation of novel auditory information is essentially the learning mechanism of linguistic or other auditory sequences. In the context of temporal prediction of acoustics, we hypothesize that successful prediction match causes faster response and failed predictions cause significant delay in response, which are likely mediated by cerebellar engagement as suggested in our data.

### 4.3. Decoding Neural Function through Visual Interpretation

CNNs are the most successful models in computer vision tasks. Although rarely used for analytical purposes due to their complexity in interpretation, these models are still promising due to their potential to model any arbitrary functions. CNNs have been successful in modeling neural events and brain-behavior relationships from EEG measures [66]. For instance, CNN and its variants have been effective in modeling cognitive load [29,33,67], seizure detection [68,69], motor imagery [70,71,72,73,74] and sleep stage scoring [75,76] from EEG recordings. Recently, the use of CNN has extended to the domain of audiology. Most notably CNNs have been used as a model for decoding speech-in-noise [77], replicating functions of auditory sensory cells [78] and solving the cocktail party problem [79]. Here, we, extend these results by demonstrating the efficacy of CNNs in decoding the acoustic-phonetic mapping inherent to speech perception.

Relevant to our study, Al-Fahad et al. combined stability selection and support vector machines (SVM) to reveal functional connectivity structures underlying fast, medium, and slow RTs [32]. Mahmud et al. decoded neural factors contributing to age-related hearing loss using the same framework [80]. These studies showed that quantitative feature importance automated by machine learning tools allows limited but useful interpretation of the relationship between neural activities and behavioral responses. However, model-agnostic feature selection techniques like stability selection are computationally expensive for CNN models. Class activation mapping tools such as CAM [81], GradCAM [31], CNN-fixation [82] and EigenCAM [83] are the only viable techniques that allow insight into the learned representation of CNN but are limited to visual depiction. Recently, GradCAM has been extended to decode neural events from EEG data [84,85,86,87] but a lack of quantification process impedes the in-depth analysis required for reaching a conclusive hypothesis. Therefore, we have laid out a process to quantify feature importance through the extraction of activation values from the class activation maps and conducted further statistical analysis to establish effect measures required to decode brain-behavior relationships.

Our contribution in the current study is not limited to modeling EEG with CNN and interpretation through activation values. Rather, we have designed a framework incorporating techniques that allow the usage of these DL tools for EEG analysis. For instance, we have adopted a randomized algorithm [29] which allowed us to use ERP samples in the DL context. The usual computation of ERPs inadvertently shrinks the number of observations and increases the bias of the dataset significantly which are impediments to ML/DL modeling. The algorithm allows us to augment ERP samples through the random combination of trials and controls the overall variance of the dataset by ensuring there is minimal overlap in these combinations. We have further adopted a composite representation of spatial-spectral features extracted from the ERP signals which allowed us to utilize the power of CNN and its interpretation tools.

### 4.4. Limitations & Future Direction

Despite the mentioned benefits, the proposed framework has some limitations worth noting. First, we acknowledge that Guided-GradCAM as a visual interpretation tool is incomplete. It is not yet clear whether Guided-GradCAM captures pixel-space feature importance learned by deep CNN models [88]. Specifically, Guided-Backpropagation [89] as a visual interpretation technique is still a matter of debate. One way to further validate our interpretation here using Guided-GradCAM is to extend this framework to other EEG studies and perceptual-cognitive paradigms beyond the relatively simple speech perception task used here.

Aside from the limitation of our analytical framework, we acknowledge that further studies must be conducted to fully understand the source of perceptual variation in CP. For instance, we did not decode temporal aspects of the neural organization driving behavioral reports. Temporal dynamics are important to understand because they explain how time differential engagement of neural regions affects behavior. Another limitation of our analysis is that we focused only on measuring the effect of individual scalp regions and did not consider the underlying neuronal sources of these activities nor possible interactions between regions (i.e., functional connectivity). Future studies could address how these factors contribute to behavioral variation in speech perception.

## 5. Conclusions

In the present study, we have conducted a novel, data-driven ERP analysis to establish neural regions as well as their oscillatory characteristics in driving behavioral decision speed (RTs) in a speech categorization task. From our analysis, it is evident that the neural sources of RT variation are encoded in α and β band oscillation over the left frontal, right prefrontal/frontal, and right cerebellar regions. These spatial-spectral correlates suggest that decision speed in categorizing speech tokens is affected by multiple processes including attention, template matching, temporal prediction of acoustics, motor control, and decision uncertainty.

We have established a generic EEG/ERP analytical framework centered on deep learning tools which could be extended to other EEG-based studies. DL models are typically avoided when the objective is an interpretation of neural events. Here, we show that learned representations of these complex models (depicted by class activation mapping tools) can be used to decode neural events of perception and behavior.

## Figures and Tables

**Figure 1 brainsci-13-00075-f001:**
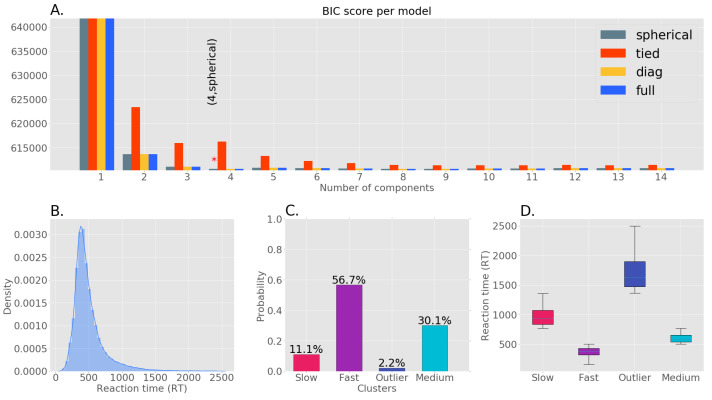
Clustering of RT data. (**A**) Bayesian Information Criterion (BIC) scores of models with different numbers of components and covariance types, the ‘*’ denotes the model with the lowest BIC score. (**B**) Original RT distribution. (**C**) The probability of each RT cluster using the GMM with the lowest BIC score. (**D**) The RT range of each cluster (slow: 772–1360 ms, fast: 100–504 ms, outlier: 1364–2500 ms, medium: 506–770 ms.

**Figure 2 brainsci-13-00075-f002:**
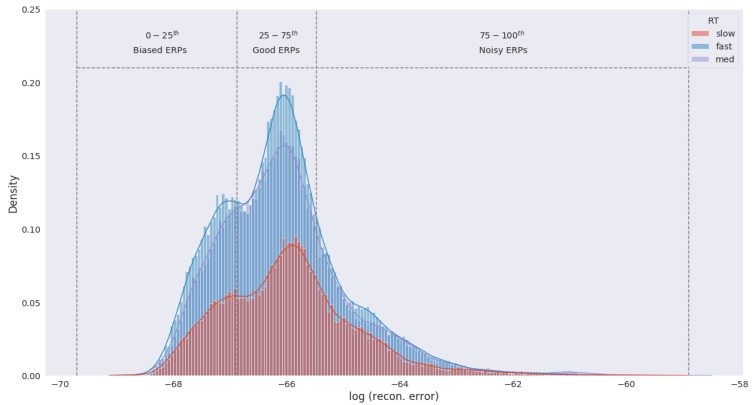
Distribution of ERP reconstruction error from the raw EEG eigenspace. Density was estimated by individual RT category through kernel density estimation (KDE). We consider ERPs within 0–25th percentile of the error distribution to be biased as these samples are far from the original EEG trials. Similarly, ERPs within 75–100th percentile are noisy due to close proximity to the raw EEG signals. ERPs within 25–75th percentile are considered samples with optimal variation and are selected for modeling RTs.

**Figure 3 brainsci-13-00075-f003:**
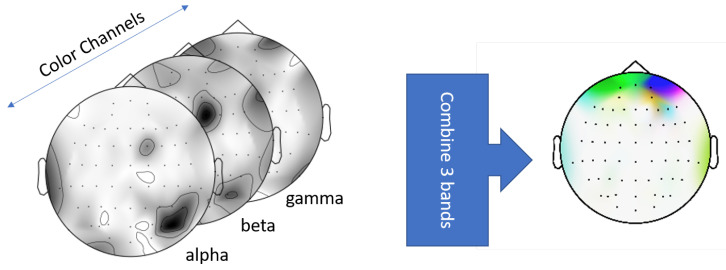
Combined bandpower topomap through stacking of the α, β, and γ topomaps across RGB channels: Red = α, Green = β. Blue = γ.

**Figure 4 brainsci-13-00075-f004:**
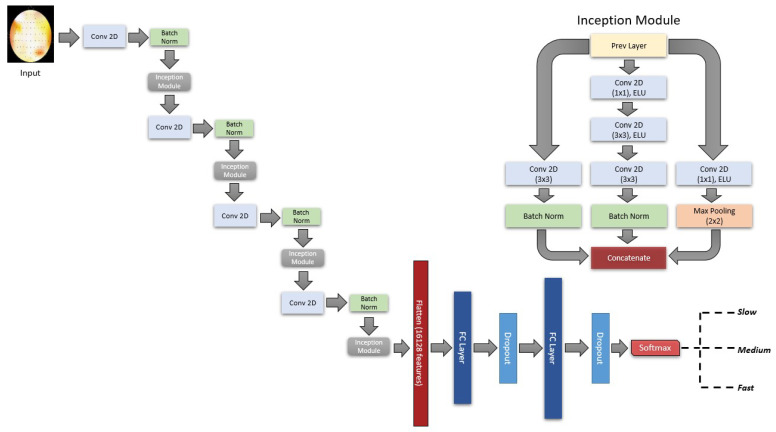
Network architecture selected by the TPE algorithm. The bandpower topomaps are the inputs of the model and are transformed through 4 consecutive convolution and inception reduction layers. The configuration of the Inception reduction layer is shown in the upper right corner. The first and interim convolution layers contain 52, 104, 156, and 208 filters with kernel size 5 × 5. The first and second FC layer contains 713 and 1019 units respectively with each of the FC layers followed by a dropout layer (rate = 0.47). The exponential linear unit (ELU) [41] function is used as activation for all the convolution and FC layers. The output layer contains 3 units with softmax activation corresponding to the 3 categorical RTs.

**Figure 5 brainsci-13-00075-f005:**
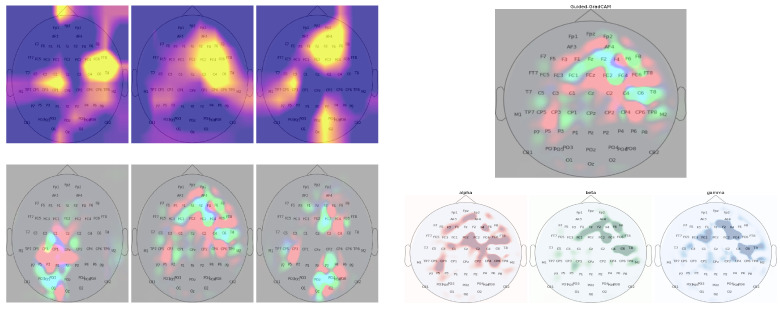
(**Left**) Examples of GradCAM activation maps (1st row) and their corresponding pixel-space feature map from Guided-GradCAM (2nd row). GradCAM activation maps show some noise as evident by activation in the background. In contrast, the pixel-space activation maps show more noise-free and detailed feature activation. (**Right**) Band-specific spatial activation from Guided-GradCAM. The red, blue, and green color channels of the pixel-space activation map correspond to the learned spatial feature importance of α, β, and γ frequency band.

**Figure 6 brainsci-13-00075-f006:**
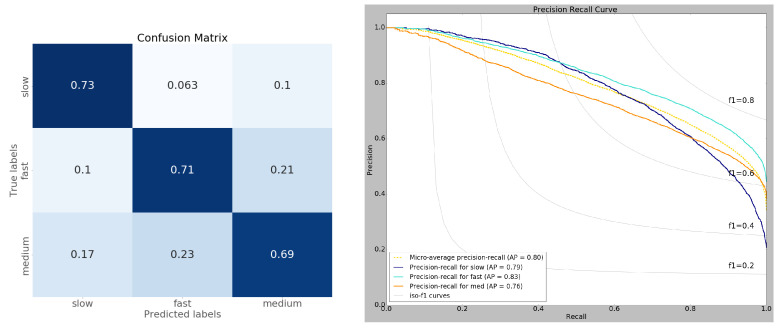
(**Left**) Normalized confusion matrix of model predicted RTs. The model makes more confusion between medium RTs with slow and fast RTs. The model also confuses some fast RTs as medium RTs. (**Right**) Precision-Recall curve of the model prediction. The model learned the spatial-spectral patterns of fast RTs (AP = 0.83) efficiently followed by slow (AP = 0.79) and medium RTs (AP = 0.76). The fast RTs are also the most stable as indicated by the f1 score (>0.6) under extreme thresholds. The increased model performance on fast RTs might be due to having the majority of samples and low uncertainty (42% of the training data).

**Figure 7 brainsci-13-00075-f007:**
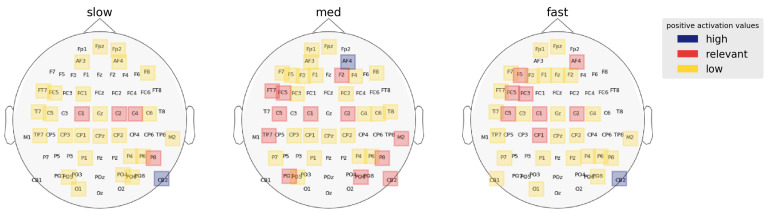
Regions of interest for each RT category as identified by the CNN model. Activation scores are averaged across the frequency bands to acquire salient regions of each RT category. Left frontal activation is inversely proportional to RTs.

**Figure 8 brainsci-13-00075-f008:**
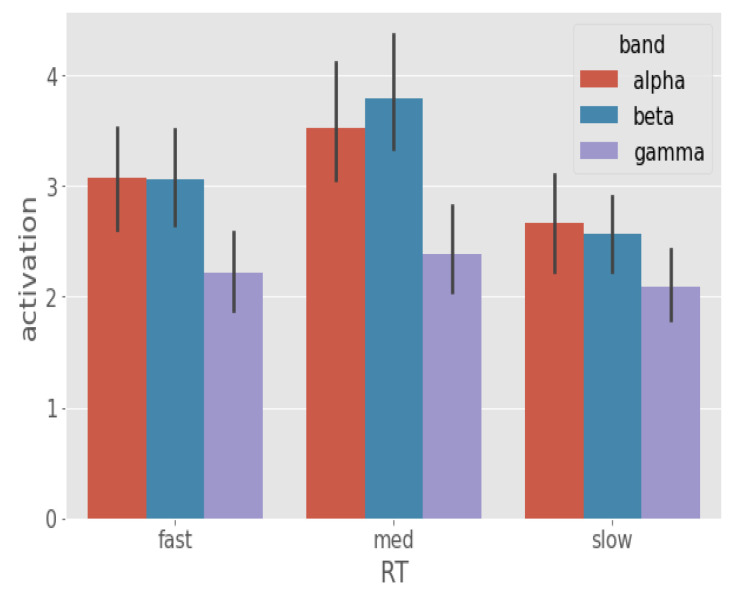
Band activation (95% CI) comparison between the RT categories. Band activation was acquired by averaging the activation scores over 64 electrodes of α, β, and γ frequency bands. From these band activations, we observe that α and β bands are the primary correlates of RT whereas the effect of γ band is minor.

**Figure 9 brainsci-13-00075-f009:**
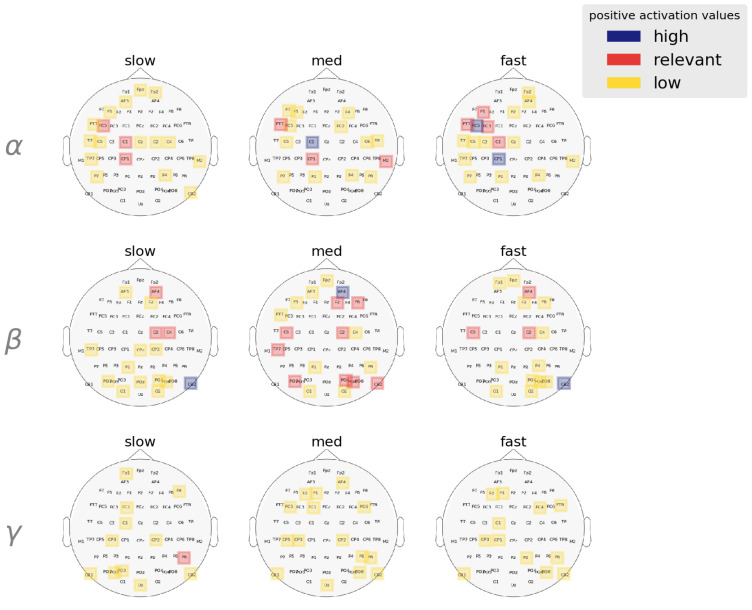
Band-wise spatial features salient to each RT category. Activation values are averaged over all trials and are clustered using GMM to determine high, relevant, and low activation ranges. α and β activation are lateralized in the left and right hemispheres, respectively. Spatial features in the γ bands are mostly deemed irrelevant by the model in describing behavioral RT speeds.

**Table 1 brainsci-13-00075-t001:** Performance metric of the best CNN model. Based on the F1 scores of the RT categories, the model more effectively learned the spatial-spectral patterns of fast and slow RT compared to medium RTs.

	Precision	Recall	F1-Score
slow	0.73	0.66	0.69
fast	0.71	0.79	0.75
med	0.69	0.64	0.66
macro avg	0.71	0.70	0.70
weighted avg	0.71	0.71	0.71

## Data Availability

Data are available upon request from the authors.

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
