# Peer review of "Unraveling Spatial-Spectral Dynamics of Speech Categorization Speed Using Convolutional Neural Networks"

_brainsci, 2022, doi:10.3390/brainsci13010075_

Round 1
Reviewer 1 Report
I have very few comments on this:
Figure 1 unnecessary
please mention what is the upper bound frequency of gamma
If your RT goes up to 1300 ms why did you epoch only up to 800 ms ?
What is the need of such complex spatial-spectral representation? Have you tried simple features and models? The motivation and comparison needs to be clearly depicted.
Reviewer 2 Report
In this paper, an experiment was conducted in which people were given to listen to sounds in the phonetic range from y to a at various frequencies. The subjects had to distribute these sounds into two categories, the response time was measured and saved for further grouping into 3 categories: fast, slow, medium. Also, with the help of measuring devices, an analysis was carried out of which part of the brain is responsible for receiving a certain type of signals.
The obtained data were preprocessed and a neural network model capable of predicting the response time was built on their basis.
All the results obtained and the directions for future work are sufficiently discussed in the article, which confirms its scientific soundness.
This paper may be accepted for publication in Mathematics journal after improving several issues:
1. Give a more strict problem statement and highlights of the paper (maybe, add this part to Introduction).
2. Could you explain how the number of subjects and their proportion (male-female) was motivated?
3. Misprints:
3.1. Line 25: delete "is" at the beginning;
3.2. Line 203: should be "Fig.4" instead of "Fig.3";
3.3 Line 236: broken citation;
3.4 Line 241: should be "Fig.6" instead of "Fig.5";
3.5 Lines 268-275 should be deleted as they are doubles;
3.6. Title of Fig. 10: should be "modeling";
3.7. Line 503: should be "behavior";
3.8. Check if the used acronyms are introduced.
